# Predictors of specialist somatic healthcare utilization among older people with intellectual disability and their age-peers in the general population: a national register study

Magnus Sandberg [ID],[1] Anna Axmon [ID],[2] Gerd Ahlström [ID],[1] Jimmie Kristensson [ID] [1,3]

[1]Department of Health Sciences, Lunds Universitet, Lund, Sweden
[2]EPI@LUND (Epidemiology, Population studies and Infrastructures at Lund University), Division of Occupational and Environmental Medicine, Lunds University, Lund, Sweden
[3]Institute for Palliative Care, Lund University, Region Skåne, Lund, Sweden

**Correspondence to**
Dr Magnus Sandberg;
Magnus.Sandberg@med.lu.se

## ABSTRACT

**Objectives** To compare somatic healthcare usage among older people with intellectual disabilities (ID) to that of their age-peers in the general population, taking into account health and demographic factors, and to identify predictors for somatic healthcare usage among older people with ID.

**Participants** Equally sized cohorts, one with people with ID and one referent cohort, one-to-one-matched by sex and year of birth, were created. Each cohort comprised 7936 people aged 55+ years at the end of 2012.

**Design** Retrospective register-based study.

**Setting** All specialist inpatient and outpatient healthcare clinics in Sweden.

**Outcome measures** Data regarding planned/unplanned and inpatient/outpatient specialist healthcare were collected from the Swedish National Patient Register for 2002–2012. Diagnoses, previous healthcare usage, sex, age and cohort affiliation was used to investigate potential impact on healthcare usage.

**Results** Compared with the referent cohort, the ID cohort were more likely to have unplanned inpatient and outpatient care but less likely to have planned outpatient care. Within the ID cohort, sex, age and previous use of healthcare predicted healthcare usage.

**Conclusions** Older people with ID seem to have lower risks of planned outpatient care compared with the general population that could not be explained by diagnoses. Potential explanations are that people with ID suffer from communication difficulties and experience the healthcare environment as unfriendly. Moreover, healthcare staff lack knowledge about the particular needs of people with ID. Altogether, this may lead to people with ID being exposed to discrimination. Although these problems are known, few interventions have been evaluated, especially related to planned outpatient care.

## STRENGTHS AND LIMITATIONS OF THIS STUDY

⇒ A strength is that the study uses data from 11 consecutive years allowing analyses to investigate healthcare usage following specific diagnoses adjusted for previous healthcare usage.
⇒ Another strength is that the study uses national registers with high validity data.
⇒ A limitation of the study is that primary healthcare data are not included.
⇒ Another limitation is that both the intellectual disabilities and the referent cohort consist of healthy survivors.

impairment,[5 6] musculoskeletal diseases[6] and thyroid conditions.[5]

Recent decades have seen an increasing life span among people with ID, resulting in an increasing number of older people with ID.[7] A consequence of this is that people with ID live longer with chronic diseases, and it is well known that older people with ID experience high rates of multimorbidity.[8 9] Thus, older people with ID would be expected to have greater healthcare needs and usage than their age-peers in the general population. Even so, they have been found to have lower rates of specialist healthcare, especially in higher ages.[10] A possible explanation for this is that healthcare is not necessarily sought at the right care level, that is, whereas older people in the general population are often referred to specialist care, older people with ID are treated in primary care. Another reason may be that some disorders are underdiagnosed/treated among older people with ID. For example, many pain and psychiatric diagnoses depend on the ability of the patient to communicate symptoms, which may be difficult for people of more severe ID.[11 12]

## BACKGROUND

A range of chronic diseases are found to a higher degree among people with intellectual disability (ID) than in the general population. These include diabetes,[1 2] cardiovascular diseases,[3] osteoporosis,[4] visual and hearing

Previous studies have reported that older people in the general population with physical as well as social unmet needs are often admitted to acute medical wards.[13 14] Thus, having a high amount of unplanned care and low amount of planned care could be an indication of unmet needs. To enable provision of proper healthcare to older people with ID, it is important to investigate factors that may predict different forms of healthcare usage.

Previous studies regarding somatic healthcare usage among people with ID have demonstrated associations between a range of demographic (eg, age, disability level), socioeconomic (eg, income, living arrangements), health (eg, subjective and self-perceived health status) and treatment (eg, polypharmacy) factors on one hand, and hospital admissions[15–19] and outpatient care[20–22] on the other. There are also studies investigating various healthcare usage patterns.[23 24] However, the majority of these studies are cross-sectional without the possibility to investigate predictors of healthcare usage. In addition, they investigate only subgroups of the ID population, for example, those at day care centres,[16] admitted for orthopaedic procedures[23] or enrolled in a home and community-based waiver.[20] In one study[17] that included both inpatient and outpatient data, outpatient visits was only measured with a dichotomised variable, more than 24 visits, or 24 or less visits. One exception is a comprehensive English study[24] investigating emergency department admissions and annual health check among 21 859 adults with ID compared with 152 846 age-matched, gender-matched and practice-matched controls without ID in 2009–2013. They found that people with ID were more likely than the general population to have emergency department admissions, and for reasons that were more likely to be deemed as preventable also after adjusting for nine different comorbidities. They also found that annual health checks for adults with ID did not have any effects on emergency department admissions compared with controls.[24] But also here, they only investigated certain healthcare types and diagnoses. Hence, there is a need for studies investigating somatic healthcare usage over a longer period.

The aims of the present study were to (1) compare somatic specialist healthcare usage among older people with ID to that of their age-peers in the general population, taking into account a range of health and demographic variables and (2) develop a model for prediction of healthcare usage among older people with ID.

## MATERIAL AND METHODS
### Study design
This was a longitudinal, retrospective, population register-based study. The study was approved by the Regional Ethical Review Board in Lund (see Ethics approval statement for further information) and performed in accordance with the Helsinki declaration.[25]

### Setting
Sweden has a welfare system mainly funded by taxes.[26] People with permanent functional impairments can receive support from the municipality according to the Act concerning Support and Service for Persons with Certain Functional Impairments (LSS).[27] The support available is regulated in the LSS act, and includes, for example, daytime activities and residential care. It also includes support to next of kin, for example, relief services. All provided support is recorded in the national LSS-register, which is maintained by the Swedish National Board of Health and Welfare. Reporting to this register is mandatory for all municipalities. The LSS act distinguishes between three types of functional impairments, whereof one is those that have been present since birth or early years. These comprise ID and autism spectrum disorders (ASD). Thus, it is possible to separate this group of people from those with intellectual functional impairment acquired in adult life, for example, caused by traumatic brain injury. People who cannot manage their day-to-day existence, have the possibility to apply for personal assistance or municipal care and/or social services according to the Social Services Act.[28] However, in Sweden, as in many other countries, informal caregivers are responsible for much of the care.

Specialist healthcare in outpatient and inpatient facilities, primary care, rehabilitation and advanced home care provided by the 21 county councils is regulated by the Health and Medical Services Act (HSL).[29] The act establishes equal access to healthcare for all Swedish residents regardless of where they live. Specialist inpatient and outpatient healthcare provided according to the HSL is registered in the National Patient Register (NPR). This register contains information on inpatient episodes and outpatient visits, with one primary and up to 21 secondary diagnoses recorded according to ICD-10 (International Statistical Classification of Diseases and related health problems 10th revision)[30] at discharge. Data are also available on whether the healthcare episode was planned (ie, if an appointment or admission was made beforehand) or unplanned. Unplanned episodes are more or less acute, and can be everything from an emergency department visit (unplanned outpatient visit) to an unplanned admission to an orthopaedic ward because of gallstone (unplanned inpatient care). The dates of hospital admission and discharge are recorded for inpatient healthcare, and the date of the visit for outpatient specialist healthcare. In this study, only data from somatic healthcare were used, but nonetheless, these could include both somatic and psychiatric diagnoses.

### Study populations
The ID cohort comprised all people aged 55 years and older who received support and social services according to the LSS act[27] during 2012, were recorded as belonging to the group of people with functional impairment since birth or early age (ie, having ID and/or ASD), and alive at the end of 2012. The cut-off at 55 years was chosen to

define 'old' because the ageing process can start earlier in people with ID.[31] A referent cohort from the general population (gPop cohort) was established through one-to-one matching by birth year and sex from the Swedish National Population Register.[32] The matching was performed by Statistics Sweden.[33] In total, 15 872 people were included in the study, 7936 in the ID cohort and an equal number in the gPop cohort. In both groups the mean age in 2012 was 65 years and 45.5% were women. However, as the data was collected retrospectively with respect to the sample, their age at the beginning of the study period in 2002 was 44 or older. If a person was included in the ID cohort, they could not be matched as a referent.

## Material

Pseudonymised inpatient and outpatient healthcare data for all individuals in the two cohorts were collected from the NPR for the 11-year period from 1 January 2002 to 31 December 2012. An individual could have several registrations for one inpatient period because of a change in ward during the hospital stay. In this paper, data on all registrations and their corresponding diagnoses are used.

## Patient and public involvement

Patients and/or the public were not involved in the design, or conduct, or reporting or dissemination plans of this research. However, representatives from a patient organisation, The Swedish National Association for People with Intellectual Disability (FUB), which is an advocacy organisation working to enable people with an ID to live a good life, has been included in the project's reference group.

## Statistical analysis

Potential impact of diagnoses, previous healthcare usage, sex, age and cohort affiliation (aim 1 only) on the four types of healthcare usage (planned/unplanned, inpatient/outpatient healthcare) was evaluated using Poisson models, thus estimating relative risks (RRs) with 95% CIs. Calendar year was used to indicate repeated measures. A similar method was employed to evaluate aim 2, with the exception that only data from the ID cohort were used and thus cohort affiliation was not included in any of the models.

Diagnoses were investigated on block level, as defined in ICD-10. Yearly inpatient healthcare usage was categorised as 0, 1, 2 or 3+ visits, whereas yearly outpatient healthcare usage was categorised as 0, 1, 2–5, 6–10 or 11+ visits. Age was aggregated to 10-year intervals.

Both aims and all four types of healthcare usage were investigated using similar procedures (figure 1): First, bivariate analyses were performed to investigate possible associations between different diagnoses and healthcare usage. Second, for each type of healthcare and aim, a multivariate model was employed, including all diagnoses with p<0.10 in the bivariate models. Third, all diagnoses with p<0.10 in the multivariate model were included in final models (one per type of healthcare and aim) which also included cohort affiliation (aim 1 only), sex, age and healthcare usage the previous year. Fourth, the predictive ability of two different models was assessed using the area under the curve.

In the final model for planned inpatient care in the ID cohort, the Hessian matrix was singular and the convergence criteria were not satisfied. We therefore ran collinearity diagnostics. This was done in a plain regression model as the GML module in SPSS does not provide this option. Diagnostic chapters were removed from the final model one by one according to their VIF-value (variance inflation factor) until the convergence criteria were satisfied. Hence, these blocks were removed due to multicollinearity: A50-A64 (Infections with a predominantly sexual mode of transmission), P10-P15 (Birth trauma), Q60-Q64 (Congenital malformations of the urinary system), B50-B64 (Protozoal diseases), G50-G59 (Nerve, nerve root and plexus disorders), K70-K77 (Diseases of liver), D00-D09 (In situ neoplasms), K65-K67 (Diseases of peritoneum), G20-G26 (Extrapyramidal and movement

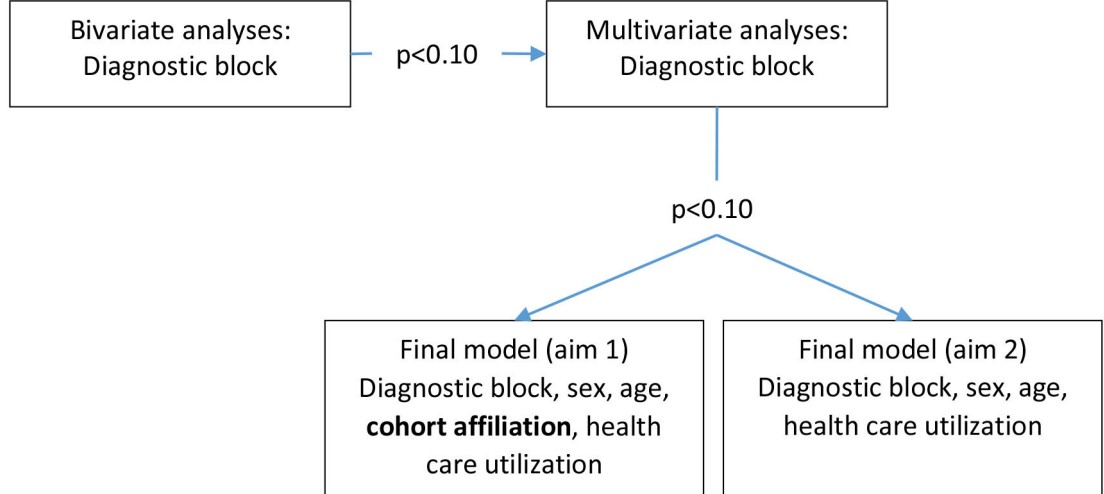

**Figure 1** Flowchart of the statistical analyses for somatic healthcare usage (unplanned inpatient care, unplanned outpatient care, planned inpatient care and planned outpatient care) the following year.

disorders), J30-J39 (Other diseases of upper respiratory tract), I80-I89 (Diseases of veins, lymphatic vessels and lymph nodes, not elsewhere classified), K50-K52 (Non-infective enteritis and colitis), J95-J99 (Other diseases of the respiratory system), M80-M94 (Osteopathies and chondropathies), R30-R39 (Symptoms and signs involving the urinary system), K80-K87 (Disorders of gallbladder, biliary tract and pancreas), K40-K46 (Hernia), G80-G83 (Cerebral palsy and other paralytic syndromes), F70-F79 (Mental retardation), M60-M79 (Soft tissue disorders), I20-I25 (Ischaemic heart diseases), M00-M25 (Arthropathies) and Z40-Z54 (Persons encountering health services for specific procedures and healthcare).

All analyses were made in IBM SPSS statistics V.23.

## RESULTS

In multivariate models including both cohorts, people in the ID cohort had increased risk of unplanned care—both inpatient and outpatient—compared with the gPop cohort (RR 1.40; 95% CI 1.33 to 1.48, and RR 1.14; 95% CI 1.08 to 1.20, respectively). However, those with ID had decreased risk of planned outpatient care (RR 0.91; 95% CI 0.89 to 0.94). There was no change in risk for planned inpatient care (RR 1.01; 95% CI 0.94 to 1.09).

Within the ID cohort, men had decreased risk of unplanned outpatient care as well as planned care (both inpatient and outpatient) compared with women (RR 0.93; 95% CI 0.88 to 0.99, RR 0.92; 95% CI 0.85 to 0.99 and RR 0.92; 95% CI 0.88 to 0.95, respectively; table 1). Increasing age was associated with increased risk of inpatient care, both planned and unplanned, but with decreased risk of planned outpatient care (table 1). Again, with only a few exceptions, healthcare the previous year carried an increased risk of healthcare usage (table 1). No diagnosis increased the risk of any type of healthcare (table 2).

**Table 1** Relative risks (RRs) with 90% CIs for somatic healthcare usage (unplanned/planned inpatient/outpatient), for sex, age and healthcare usage the year before, within a cohort of 7936 older people with intellectual disability

|  | Unplanned | | Planned | |
|---|---|---|---|---|
|  | Inpatient | Outpatient | Inpatient | Outpatient |
|  | RR (90% CI) | RR (90% CI) | RR (90% CI) | RR (90% CI) |
| Men vs women | 1.04 (0.99 to 1.08) | **0.93 (0.88 to 0.99)** | **0.92 (0.85 to 0.99)** | **0.92 (0.88 to 0.95)** |
| Age (years) |  |  |  |  |
| 50–59 vs <50 | **1.30 (1.18 to 1.44)** | 1.12 (0.97 to 1.28) | **1.28 (1.05 to 1.55)** | **0.92 (0.86 to 0.99)** |
| 60–69 vs <50 | **1.50 (1.35 to 1.66)** | 1.03 (0.90 to 1.19) | **1.48 (1.21 to 1.80)** | **0.79 (0.73 to 0.85)** |
| 70–79 vs <50 | **1.75 (1.57 to 1.96)** | 0.91 (0.77 to 1.07) | **1.56 (1.26 to 1.93)** | **0.75 (0.69 to 0.82)** |
| 80–89 vs <50 | **2.07 (1.81 to 2.37)** | 0.96 (0.75 to 1.24) | 0.97 (0.69 to 1.39) | **0.54 (0.45 to 0.64)** |
| 90+ vs <50 | **2.33 (1.60 to 3.41)** | 1.12 (0.51 to 2.45) |  | 0.32 (0.08 to 1.24) |
| Planned inpatient previous year |  |  |  |  |
| 1 vs 0 | **1.14 (1.04 to 1.24)** | 1.05 (0.95 to 1.16) | **1.90 (1.64 to 2.21)** | 1.00 (0.95 to 1.06) |
| 2 vs 0 | 1.07 (0.89 to 1.29) | 1.02 (0.83 to 1.26) | **2.89 (2.22 to 3.75)** | 1.06 (0.94 to 1.20) |
| 3+ vs 0 | **1.43 (1.09 to 1.88)** | 0.99 (0.68 to 1.45) | **4.03 (2.84 to 5.70)** | **1.34 (1.11 to 1.61)** |
| Planned outpatient previous year |  |  |  |  |
| 1 vs 0 | **1.25 (1.19 to 1.33)** | **1.10 (1.02 to 1.18)** | **2.07 (1.86 to 2.30)** | **1.89 (1.80 to 1.99)** |
| 2–5 vs 0 | **1.40 (1.31 to 1.50)** | **1.34 (1.23 to 1.45)** | **2.77 (2.46 to 3.13)** | **2.59 (2.45 to 2.73)** |
| 6–10 vs 0 | 1.23 (0.97 to 1.56) | 0.94 (0.72 to 1.22) | **2.66 (1.83 to 3.87)** | **2.87 (2.59 to 3.18)** |
| 11+ vs 0 | 1.36 (0.99 to 1.88) | **1.53 (1.11 to 2.11)** | **3.80 (2.51 to 5.76)** | **2.45 (2.05 to 2.93)** |
| Unplanned inpatient previous year |  |  |  |  |
| 1 vs 0 | **2.02 (1.89 to 2.16)** | **1.14 (1.06 to 1.23)** | **1.59 (1.42 to 1.78)** | **1.04 (1.00 to 1.09)** |
| 2 vs 0 | **2.38 (2.17 to 2.62)** | **1.20 (1.08 to 1.34)** | **1.78 (1.49 to 2.14)** | **1.08 (1.00 to 1.16)** |
| 3+ vs 0 | **2.70 (2.39 to 3.04)** | **1.19 (1.05 to 1.35)** | **2.43 (2.03 to 2.90)** | **1.16 (1.07 to 1.26)** |
| Unplanned outpatient previous year |  |  |  |  |
| 1 vs 0 | **1.25 (1.17 to 1.33)** | **1.78 (1.65 to 1.91)** | **1.20 (1.06 to 1.35)** | 1.02 (0.97 to 1.07) |
| 2–5 vs 0 | **1.42 (1.29 to 1.56)** | **2.73 (2.47 to 3.02)** | **1.21 (1.00 to 1.46)** | **1.15 (1.08 to 1.23)** |
| 6–10 vs 0 | **1.33 (1.01 to 1.74)** | **3.66 (3.05 to 4.39)** | 1.18 (0.69 to 2.00) | 0.97 (0.78 to 1.20) |
| 11+ vs 0 | **1.52 (1.09 to 2.13)** | **3.74 (2.90 to 4.82)** | **2.22 (1.48 to 3.34)** | **1.45 (1.11 to 1.89)** |

All analyses are based on multivariate models adjusted for relevant diagnoses on International Statistical Classification of Diseases 10th revision block-level. Bold indicates p<0.1.

**Table 2** Relative risks for somatic healthcare usage (unplanned/planned inpatient/outpatient) for people with diagnosis (A20-A29, A30-A49, etc) compared with those without, within a cohort of 7936 older people with intellectual disability

| | Unplanned | | Planned | |
|---|---|---|---|---|
| | Inpatient | Outpatient | Inpatient | Outpatient |
| A20-A28: Certain zoonotic bacterial diseases | | | | 2.25 |
| A30-A49: Other bacterial diseases | | | | |
| A50-A64: Infections with a predominantly sexual mode of transmission | 10.78 | | | 2.83 |
| A90-A99: Arthropod-borne viral fevers and viral haemorrhagic fevers | | | | 1.45 |
| B20-B24: HIV disease | 2.21 | | | |
| B25-B34: Other viral diseases | | | | |
| B50-B64: Protozoal diseases | 4.29 | | | |
| B95-B98: Bacterial, viral and other infectious agents | | 0.68 | | |
| B99-B99: Other infectious diseases | | | | 0.69 |
| C00-C97: Malignant neoplasms | 0.77 | | | 1.43 |
| D00-D09: In situ neoplasms | 2.18 | | | |
| D80-D89: Certain disorders involving the immune mechanism | | | | 1.72 |
| E10-E14: Diabetes mellitus | 1.27 | 1.18 | | 1.39 |
| E15-E16: Other disorders of glucose regulation and pancreatic internal secretion | | 2.59 | | |
| E70-E90: Metabolic disorders | 0.79 | 0.73 | | |
| F00-F09: Organic, including symptomatic, mental disorders | | | | 0.79 |
| F10-F19: Mental and behavioural disorders due to psychoactive substance use | 1.56 | 1.55 | | |
| F40-F48: Neurotic, stress-related and somatoform disorders | | 1.28 | | |
| F50-F59: Behavioural syndromes associated with physiological disturbances and physical factors | 1.78 | | | |
| F70-F79: Mental retardation | | 0.91 | | 0.82 |
| F80-F89: Disorders of psychological development | | 0.91 | | 0.81 |
| F99-F99: Unspecified mental disorder | | | | |
| G20-G26: Extrapyramidal and movement disorders | 1.32 | 1.38 | | 1.36 |
| G40-G47: Episodic and paroxysmal disorders | 1.62 | 1.53 | | 1.49 |
| G60-G64: Polyneuropathies and other disorders of the peripheral nervous system | 1.84 | | | |
| G70-G73: Diseases of myoneural junction and muscle | | 1.93 | | |
| G80-G83: Cerebral palsy and other paralytic syndromes | 1.31 | | | |
| H15-H22: Disorders of sclera, cornea, iris and ciliary body | | 1.37 | | |
| H25-H28: Disorders of lens | | 0.81 | 1.37 | |
| H30-H36: Disorders of choroid and retina | | | | 1.19 |
| H40-H42: Glaucoma | | | | 1.61 |
| H65-H75: Diseases of middle ear and mastoid | | | | 1.62 |
| H90-H95: Other disorders of ear | | 1.37 | | |
| I10-I15: Hypertensive diseases | | | | |
| I20-I25: Ischaemic heart diseases | 1.17 | | | 1.12 |
| I30-I52: Other forms of heart disease | | | | 1.17 |
| I60-I69: Cerebrovascular diseases | | | | |
| I80-I89: Diseases of veins, lymphatic vessels and lymph nodes, not elsewhere classified | | | | 0.83 |
| J00-J06: Acute upper respiratory infections | 1.28 | | | 0.77 |
| J09-J18: Influenza and pneumonia | | | | 0.84 |
| J20-J22: Other acute lower respiratory infections | | | | |
| J30-J39: Other diseases of upper respiratory tract | | | | 1.28 |
| J40-J47: Chronic lower respiratory diseases | 1.19 | | | |

Continued

**Table 2**  Continued

|  | Unplanned | | Planned | |
| --- | --- | --- | --- | --- |
|  | Inpatient | Outpatient | Inpatient | Outpatient |
| J60–J70: Lung diseases due to external agents |  |  |  | 0.78 |
| J95–J99: Other diseases of the respiratory system |  |  |  |  |
| K40–K46: Hernia |  |  |  |  |
| K50–K52: Non-infective enteritis and colitis |  | 1.42 |  | 1.44 |
| K55–K63: Other diseases of intestines | 1.22 | 1.17 | 1.37 |  |
| K65–K67: Diseases of peritoneum |  |  |  |  |
| K70–K77: Diseases of liver | 1.35 |  |  | 1.46 |
| L10–L14: Bullous disorders |  | 2.70 |  |  |
| L20–L30: Dermatitis and eczema | 1.46 | 1.66 |  |  |
| L50–L54: Urticaria and erythema | 1.98 |  |  |  |
| L80–L99: Other disorders of the skin and subcutaneous tissue | 0.64 |  |  |  |
| M00–M25: Arthropathies | 1.13 |  |  | 1.22 |
| M95–M99: Other disorders of the musculoskeletal system and connective tissue |  | 2.39 |  |  |
| N00–N08: Glomerular diseases |  |  |  | 1.51 |
| N10–N16: Renal tubulointerstitial diseases |  |  |  | 0.72 |
| N17–N19: Renal failure |  |  |  | 1.16 |
| N30–N39: Other diseases of urinary system | 1.04 |  |  |  |
| N40–N51: Diseases of male genital organs | 0.54 |  |  |  |
| P20–P29: Respiratory and cardiovascular disorders specific to the perinatal period | 0.63 |  |  |  |
| Q20–Q28: Congenital malformations of the circulatory system | 1.59 |  |  |  |
| Q60–Q64: Congenital malformations of the urinary system |  |  |  | 1.50 |
| Q90–Q99: Chromosomal abnormalities, not elsewhere classified |  | 0.84 |  |  |
| R00–R09: Symptoms and signs involving the circulatory and respiratory systems | 1.29 | 1.50 |  |  |
| R10–R19: Symptoms and signs involving the digestive system and abdomen | 1.22 | 1.22 |  |  |
| R50–R69: General symptoms and signs |  |  |  |  |
| S00–S09: Injuries to the head | 1.16 |  |  |  |
| S30–S39: Injuries to the abdomen, lower back, lumbar spine and pelvis |  |  |  |  |
| S60–S69: Injuries to the wrist and hand |  | 1.18 |  |  |
| S80–S89: Injuries to the knee and lower leg |  | 0.84 |  |  |
| T15–T19: Effects of foreign body entering through natural orifice | 1.31 |  |  |  |
| T51–T65: Toxic effects of substances chiefly non-medicinal as to source | 2.00 |  |  |  |
| Z70–Z76: Persons encountering health services in other circumstances |  | 1.29 |  |  |

All analyses are based on multivariate models adjusted for sex, age and healthcare usage previous year. Only results with p<0.1 are included.

All models, that is, based on ID cohort only, and for all four types of care, were able to predict healthcare usage better than chance (figure 2).

## DISCUSSION

Even when adjusting for demographic factors, all recorded diagnoses, and previous healthcare usage, ID was associated with increased unplanned healthcare usage. If unplanned healthcare is a measure of unmet healthcare needs, this is cause for concern. Another important finding was the lower healthcare usage among men compared with women among people with ID, even when all recorded diagnoses were taken into account. This suggests gender differences in access to healthcare among older people with ID.

The major strength of the present study is the use of national registers with high validity to identify people with ID as well as healthcare usage and diagnoses. In using data from 11 consecutive years and repeated measures analyses, we were able to investigate healthcare usage following specific diagnoses, rather than mere associations (ie, non-directional), as well as adjust for previous

**Unplanned inpatient care**

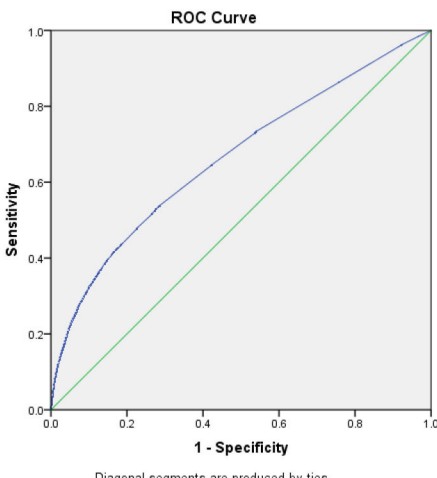

**Unplanned outpatient care**

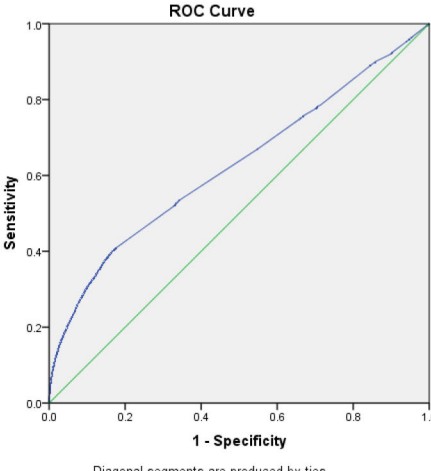

**Planned inpatient care**

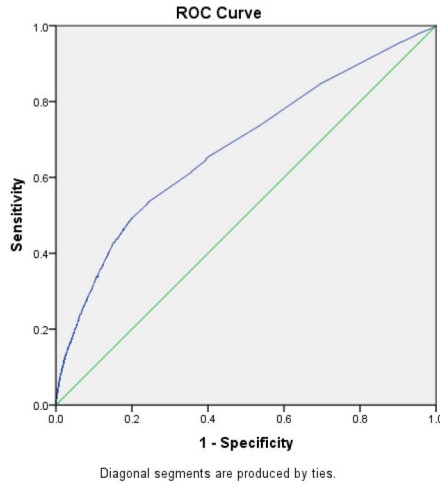

**Planned outpatient care**

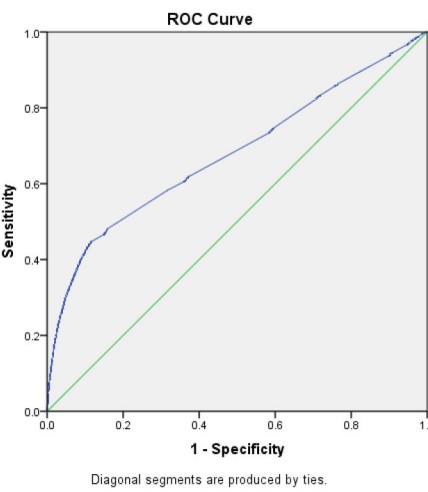

**Figure 2** ROC-curves for multivariate models predicting four types of somatic specialist healthcare usage among 7936 older people with intellectual disability, using data on age, sex, previous healthcare usage and relevant diagnoses. ROC, Receiver Operating Characteristic.

healthcare usage. Thus, the results presented for demographic factors and diagnoses are most likely not inflated by people with particularly high healthcare consumption. In addition, reporting of inpatient and specialist outpatient care to NPR is mandatory for healthcare providers and the register has been shown to have high validity.[34]

A potential weakness with the present study is the use of an administrative ID cohort. The LSS register comprises all people who have received service and support aimed at people with ID and/or ASD, including support to informal caregivers. However, those who did not receive such support in 2012 were not included in the ID cohort. This may include, for example, people with ID who are being taken care of solely by parents or other family without support from the municipality. Even so, the Swedish welfare system is designed so that vulnerable

groups are cared for by the state, county or municipality. It is rare that informal caregivers within the family do not receive any support at all. In addition, considering the age group studied, we believe that the number of people with ID being cared for by their parents is low. Another limitation with the LSS register is that the level of ID is not registered. We have therefore not been able to take this into account. We have previously used the level of ID registered in the NPR, however such analyses excluded those without such a diagnosis during the study period, including those without any healthcare usage.[35] As the data were collected retrospectively for people alive at the end of the study period, another potential weakness could be cohort effects, that is, that people with ID that survived and turned 55 years or older might be healthier than their younger non-surviving peers.

We collected information on healthcare usage as well as diagnoses from the NPR. Although this register has national coverage of specialist care, it does not include visits to primary care. Thus, even though we have good coverage of specialist healthcare usage, it is difficult to interpret the results in terms of unmet *overall* healthcare needs. As people with ID are less likely than those in the general population to get referrals from primary to specialist care[36] any bias introduced from the lack of primary healthcare data should have caused an underestimation of the higher healthcare usage associated with having ID. In addition, the diagnoses are aggregated over an 11-year period and no consideration was taken to multimorbidity. To get a deeper understanding of the determinants of unplanned healthcare and the adequacy of the care, a limited number of diagnoses or combinations of diagnoses, should be investigated more in detail in future studies.

One of our main findings is that people with ID have less planned outpatient care than the general population, with lower risks in higher ages, which is not explained by differences in diagnoses. This is surprising as people with ID have a more complex disease panorama than the general population,[4–6 9] and as the prevalence of multiple chronic diseases is a well-known explanation for higher amount of healthcare usage in the general older population.[37] In addition, it is reasonable to assume that unplanned care in terms of hospital admissions are followed-up with planned forms of care and services in order to prevent readmission.[38] Previous research has also found that annual health checks seem to reduce preventable emergency admissions.[24] Only a few studies have investigated healthcare use among people with ID compared with that in the general population.[39–44] Of these, only Skorpen, *et al*[43] reported results consistent with the present study, with adults with ID being hospitalised more frequently at younger age and less frequently at older ages compared with the general population. Most studies did not consider the influence of age on healthcare usage.[39 40 42 44] However, Hsu, *et al*[41] investigated ambulatory care visits among older people (65 years and older) with and without ID, and found that people with ID had a higher mean number of visits and mean annual visits, and that age group affected the annual visits with middle–elderly (74–84) in both groups having the highest frequency of visits. There is a range of possible explanations for the lower rates of planned healthcare among people with ID. These include difficulties accessing the healthcare system[45–47] and lower rates of planned preventive healthcare, such as breast and cervical cancer screening.[48 49] Furthermore, even when having entered the healthcare system, people with ID may have communication difficulties and experience the healthcare environment as unfriendly.[50–52] An explanation of particular concern is that the ID population is discriminated by the healthcare system.[53]

In addition to being discriminated in relation to the general population, there could also be discrimination between different groups within the ID population. The present study also revealed that women with ID have higher healthcare usage than men with ID. Gender differences in psychiatric diagnoses have been reported,[54] but do not seem to have been investigated when it comes to somatic healthcare usage. Thus, it is important to investigate whether men with ID are at risk for a 'double discrimination'.

Even though the need for interventions regarding requirement of knowledge about and various aspects of supporting people with ID have been addressed since the 1990s,[55] there is a lack of interventions studies and the reduced access to healthcare among people with ID still persists today.[50 56] In addition, adults with ID have been reported to be at high risk for preventable unplanned admissions, despite having similar primary care usage before admission as the general population.[57] Hence, further interventions are needed, for example, such that focus on communication, knowledge/information and profession-specific needs.[55] This may include enhancements of medical and nursing schools curriculums[58–60] to increase knowledge about people with ID, which may facilitate access to healthcare, which in turn may support the health and well-being among people with ID. Moreover, interventions aimed at primary care providers may be relevant, as these often are the gate keepers to other forms of care and often have continuous contact with the person with ID, and thus, have the possibility to coordinate care, and also to provide disease prevention, early detection and appropriate management.[61] People with ID in primary care have been reported to have lower continuity of care and be less likely to have longer appointments than the general population, which could be areas of improvement,[62] especially as higher continuity of care has been found to be associated to the emergency department visits among people with ID.[63] Even though initiatives have been taken to increase the number of people with ID that seek primary healthcare,[58 64 65] the effects of these interventions do not seem to have been evaluated. Furthermore, few studies have evaluated continuity of care interventions for people with ID and none with unplanned admissions as primary outcome.[66] Thus, this needs to be addressed in future research.

## Conclusions

People with ID have less planned outpatient healthcare than the general population. This difference was not explained by diagnoses, demographic factors or previous healthcare usage. Thus, it is likely a result of other factors, for example, communication difficulties and experiences of the healthcare environment as unfriendly among people with ID, as well as a lack of knowledge about ID among healthcare staff and discrimination of people with ID. Discrimination could also be the reason for less planned and more unplanned inpatient and outpatient specialist care among those with ID. These problems have been addressed for a long time, but there are still few intervention studies in the area and especially studies

targeting the problems related to planned healthcare identified in the present study. Thus, there is a great need for evaluation of healthcare interventions, especially in planned outpatient care such as primary care, which aim to give people with ID access to healthcare to the same extent as the general population.

**Acknowledgements** We would like to acknowledge FORTE – The Swedish Research Council for Health, Working Life and Welfare (grant no. 2014–4753) and the Faculty of Medicine, Lund University, Sweden, for financing this study.

**Contributors** MS participated in the design of the study and in the collection of data, interpretation of data and drafted the manuscript. GA participated in the design of the study, helped in the interpretation of the data and drafting of the manuscript and is guarantor of the study. AA participated in the design of the study, performed the analysis, interpretation of data and helped in the drafting of the manuscript. JK participated in the design of the study and helped in the interpretation of the data and drafting the manuscript. All authors read and approved the final manuscript.

**Funding** This work was funded by Forte, the Swedish Research Council for Health, Working Life and Welfare (http://forte.se/en) no. 2014-4753 with GA as principal investigator. The funders had no role in study design, data collection and analysis, decision to publish or preparation of the manuscript.

**Competing interests** None declared.

**ORCID iDs**
Magnus Sandberg http://orcid.org/0000-0002-8915-8730
Anna Axmon http://orcid.org/0000-0002-4539-8337
Gerd Ahlström http://orcid.org/0000-0001-6230-7583
Jimmie Kristensson http://orcid.org/0000-0002-3659-7860

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
