## [Reviewer comments · BMJ Open]

ARTICLE DETAILS

TITLE (PROVISIONAL)	Predictors of specialist somatic healthcare utilization among older people with intellectual disability and their age-peers in the general population: A national register study
AUTHORS	Sandberg, Magnus; Axmon, Anna; Ahlström, Gerd; Kristensson, Jimmie

VERSION 1 – REVIEW

REVIEWER	McGrath, Ryan North Dakota State University
REVIEW RETURNED	13-Mar-2023

GENERAL COMMENTS	BMJOPEN-2023-072679 presents results for predictors of healthcare utilization for persons with ID. While some parts of this paper were interesting, other areas could be improved. • In text citations should be formatted like “.5”?• Line 60: Delete “e.g.”• Statistical Analysis: Given the presentation of the study Aims, it is unclear how the current analysis will compare somatic specialist healthcare utilization among persons with ID to their age-peers in the general population. The current analyses include Poisson models (which are generally used for count data), and GLM. Perhaps a matched cohort analysis of sort should be used to compare ID to non-ID matched on age instead?• Lines 186-187: SPSS perhaps is not adequate for this analysis?• The findings in Table 1 are not striking.• Table 2 is large and dense with words. Is there another presentation of this information that could help to improve?• Make any changes to the abstract that align with those made from the text.
---

REVIEWER	Imaiso, Junko Kanazawa Univ
REVIEW RETURNED	17-Mar-2023

GENERAL COMMENTS	Thank you so much for your giving me an opportunity to review the significant research manuscript. I understood that the current study, longitudinal and retrospective study, was conducted in order to pursue the effective way of specialist somatic healthcare utilization for the older people (55-years-old and over) with ID to prevent chronic diseases in Sweden. In increasing aging society worldwide, the current study was globally significant because aging would be related to population health in a community, I thought. However, as an original research article, the validity of the results seemed not to be cleared based on the research purpose.
--

	Therefore, I would like to ask the authors to reconfirm the following points, I report to you. 1. Material and Methods 1) It seemed not to be considered about ID level of the study populations. I thought why. 2) In the current study, it was difficult for me to understand what were the unplanned and the planned inpatient or outpatient in the current study. 3) The current study seemed to be focused on individual aspects such as age, diagnosed chronic disease. However, it was difficult for me to understand the reason. Because I think that the specialist somatic healthcare utilization in a municipality may be influenced not only in the individual character, but also the environment such as family environment or residential environment. And, I wondered if in Sweden, the social supports related to specialist somatic healthcare utilization in each municipality might be equal or not. Based on the global standard, it might be easier for readers to understand if health care system or health policy in Sweden related to specialist somatic healthcare utilization for the older people with ID is explained more in detail, I thought. 2. Results 1) The basic individual characteristics of 7936 participants in the current study were needed to be showed, I thought. 2) I understood that the age of participants in the current study was 55 years old and over by the description in Study populations (line 137, page 7). However, it was seemed to be 50 years old and over according Table 1 (page 22). I had a question. 3) Table 1 and Table 2 The each title seemed not to be cleared. And, according statistical analysis (page 8-9) based on the purpose in the current study, Table 1 and Table 2 should be showed more in detail, I thought. According to the main comments, I think that the decision is major revision. If the comments are some useful for improving the research paper, I feel happy. If there are some troubles, please let me know.
--	---

VERSION 1 – AUTHOR RESPONSE

Comments from Reviewer 1:

In text citations should be formatted like “.5”?	Thank you for pointing this out. We have now changed the in-text citation at line 315.
Line 60: Delete “e.g.”	This has been revised.
Statistical Analysis: Given the presentation of the study Aims, it is unclear how the current	Thank you for this comment. You are correct in that we have used Poisson models. This has

analysis will compare somatic specialist healthcare utilization among persons with ID to their age-peers in the general population. The current analyses include Poisson models (which are generally used for count data), and GLM. Perhaps a matched cohort analysis of sort should be used to compare ID to non-ID matched on age instead?	now been added to the analyses paragraph. But such analyses estimate risk ratios for dichotomous outcomes and can be performed in SPSS.
Lines 186-187: SPSS perhaps is not adequate for this analysis?	We are aware of that there is other statistical software that other prefer for this kind of data and for this kind of analysis. But SPSS has been demonstrated to work absolutely fine for the analysis performed.
The findings in Table 1 are not striking.	We agree that the findings to some extent were expected. However, as this study is unique in studying a large sample over time, including a general population cohort, it still contributes important knowledge. Especially when it comes to planned outpatient care.
Table 2 is large and dense with words. Is there another presentation of this information that could help to improve? Make any changes to the abstract that align with those made from the text.	We agree with the reviewer. This Table has been a great challenge and we have tried different ways of presenting, but where this version was believed to be the least dense and still with all necessary details. We considered removing the name of each ICD-10 block and replacing this with the name of the ICD-chapter.

	But as ICD-10 blocks might not be common knowledge we found this solution to be more suboptimal. No changes that affect the abstract have been made.
--	--

Comments from Reviewer 2:

Material and Methods	
1) It seemed not to be considered about ID level of the study populations. I thought why.	In the LSS register the level of ID is not registered. In some cases this has been registered in the National Patient Register, but only among a minor part of the sample, and only among the proportion that had any healthcare utilization. This matter has been added in the discussion paragraph (lines 261-264)
2) In the current study, it was difficult for me to understand what were the unplanned and the planned inpatient or outpatient in the current study.	We could agree on that this could be difficult to understand and have therefor clarified this in the method paragraph (lines 137-139)
3) The current study seemed to be focused on individual aspects such as age, diagnosed chronic disease. However, it was difficult for me to understand the reason. Because I think that the specialist somatic healthcare utilization in a municipality may be influenced not only in the individual character, but also the environment such as family environment or residential	Thank you for a valuable comment. We agree that healthcare utilization depends on many different variables, including individual, but also those on societal and organizational levels. We have added a section in the method paragraph (lines 124-127 and 130-131) to better explain the Swedish context. Please note that somatic care is provided by the county council and not

environment. And, I wondered if in Sweden, the social supports related to specialist somatic healthcare utilization in each municipality might be equal or not. Based on the global standard, it might be easier for readers to understand if health care system or health policy in Sweden related to specialist somatic healthcare utilization for the older people with ID is explained more in detail, I thought.	by the municipality. There have been reports that there are some differences between the county councils, but these will probably affect both the ID and the general population cohort to the same extent.
Results	
4) The basic individual characteristics of 7936 participants in the current study were needed to be showed, I thought.	The amount of individual characteristics was restricted by the Regional Ethical review board in Lund. We only got permission to age and gender variables, and these are now presented in the results section (lines 153-155).
5) I understood that the age of participants in the current study was 55 years old and over by the description in Study populations (line 137, page 7). However, it was seemed to be 50 years old and over according Table 1 (page 22). I had a question.	Thank you for this comment. You are absolutely right that the inclusion criterion was 55+ at the end of 2012. But as this register study used retrospective data for eleven years, their age could be eleven years lower than 55. This has been clarified in the method paragraph (lines 153-155)
6) Table 1 and Table 2 The each title seemed not to be cleared. And, according statistical analysis (page 8-9) based on the purpose in the current study, Table 1 and Table 2 should be showed more in detail, I thought.	We are not sure that we fully understand your comment and in what way the titles are unclear and what details are missing in the tables. As pointed out by reviewer #1, especially Table 2 is large and dense with words, so we do not wish to add additional information. "Still, we have tried to clarify the titles both in Table 1 and 2. We hope that this